# PRR14 overexpression promotes cell growth, epithelial to mesenchymal transition and metastasis of colon cancer via the AKT pathway

Fangfang Li[1,2], Chundong Zhang[3], Lijuan Fu[1,4]*

1 Joint International Research Laboratory of Reproduction and Development of the Ministry of Education, Department of Reproductive Biology, Chongqing Medical University, Chongqing, China, 2 Medical Research Center, Southwest Hospital, Third Military Medical University (Army Medical University), Chongqing, China, 3 Department of Biochemistry and Molecular Biology, Chongqing Medical University, Chongqing, China, 4 Department of Traditional Chinese Medicine, Chongqing Medical University, Chongqing, China

* fulijuan@cqmu.edu.cn

## Abstract

**Data Availability Statement:** All relevant data are within the paper and its Supporting Information files.

### Background

PRR14 (Proline rich protein 14) was firstly identified for its ability to specify and localize heterochromatin during cell cycle progression. Aberrant expression of PRR14 is associated with the tumorigenesis and progression of lung cancer. However, its involvement in colon cancer remains unknown. Herein, we report the role of PRR14 in colon cancer.

### Methods

Colon cancer tissue microarray was used to analyze and compare the expression of PRR14 among some clinicopathological characteristics of colon cancer. HCT116 and RKO cells were transfected with siRNA to downregulate PRR14 expression. The roles of PRR14 in proliferation, migration and invasion of the cell lines were determined using cell counting kit-8, colony formation assay, wound healing assay and transwell assays respectively. The expression of PRR14 was measured using immunofluorescence, qRT-PCR and western blot. Epithelial-mesenchymal transition (EMT) markers were determined by western blot.

### Results

PRR14 was highly expressed in colon cancer tissues, and the expression level was correlated with tumor size, distant metastasis and Tumor Node Metastasis stages. Functional study revealed that downregulation of PRR14 inhibited colon cancer cells growth, migration and invasion. Furthermore, knockdown of PRR14 inhibited epithelial-mesenchymal transition (EMT) process, cell cycle-associated proteins expression and p-AKT level.

**Funding:** This work was supported by the Basic application general project of Southwest Hospital of Third Military Medical University (SWH2016JCYB-52) and Chongqing Natural Science Foundation (cstc2016jcyj0247). The funders had no role in study design, data collection and analysis, decision to publish, or preparation of the manuscript.

**Competing interests:** The authors have declared that no competing interests exist.

## Conclusion

PRR14 may promote the progression and metastasis of colon cancer, and may be a novel prognostic and therapeutic marker for the disease.

## Introduction

Colon cancer is one of the most aggressive digestive system cancers worldwide; and its incidence rate in China has been increasing dramatically over the years [1, 2]. Due to its distant metastasis and high recurrence rate, colon cancer patients often have a poor prognosis. Although a large number of genes associated with the metastasis and recurrence of colon cancer have been identified, the molecular pathogenesis of this disease is still far from being fully elucidated. Therefore, identification of new genes associated with colon cancer progression is of great importance for the prevention, early and accurate diagnosis, and treatment of the disease.

PRR14 (Proline Rich Protein 14) is a member of the proline-rich protein family. It contains one proline-rich region flanked by N- and C-terminal nuclear localization signals [3]. The proline rich domain participates in multiple protein interactions by binding to various domains [4, 5]. During cell cycle progression, PRR14 presents a dynamic distribution with a periodic disintegration and reconstruction of the nuclear lamina, and its function is to target the HP1-labeled heterochromatin to locate in the nuclear lamina during mitotic exit [3, 6]. In skeletal muscles, PRR14 promotes myogenesis via the maintenance of the nuclear lamina structure and the stimulation of the activity of MyoD [7].

Interestingly, PRR14 has been implicated in the development of tumors and cancers. PRR14 is amplified and aberrantly overexpressed in lung cancer, and promotes lung cancer cells proliferation through the activation of the PI3K/AKT/mTOR signaling pathway. Its proline-rich peptide alone can activate the PI3K pathway to promote the proliferation of lung cancer cells *in vitro* and tumorigenesis *in vivo* [8]. The expression of PRR14 in a variety of tumors in the TCGA database has also been analyzed, and it has been found that PRR14 is commonly elevated in a variety of tumors, including colon cancer [9]. Besides, it has been reported that PRR14 can be used as an independent prognostic marker and may be a potential therapeutic target in the treatment of non-small cell lung cancer (NSCLC) [10]. PRR14 promotes the growth of breast cancer cells by regulating the Ras pathway [11]. To date, the role of PRR14 in colon cancer has not been reported.

In this study, we have investigated the possible role of PRR14 in colon cancer, by which we explored PRR14 expression in colon cancer, and studied the effects of its knockdown on the proliferation, migration and invasion of colon cancer cells, as well the formation of pseudopodia.

## Materials and methods

### Cell line and cell culture

Human colon cancer cell lines, HCT116 and RKO, were purchased from ATCC. The cell lines were maintained in DMEM high glucose medium with 10% FBS (Invitrogen,USA) and penicillin (100 IU/ml) /streptomycin (100 mg/ml). Cells were incubated at 37 °C in a humidified atmosphere of 5% $CO_2$.

## Tissue array and immunohistochemistry

PRR14 protein expression was detected on a colon cancer tissue array slide from Shanghai Outdo Biotech CO. (HColA160CS01, Shanghai, China). The tissue array contained human colon cancer tissues and their corresponding adjacent normal tissues (80 cases). The primary antibody used was rabbit anti-PRR14 antibody(1:2000, HPA060265, Sigma, USA). IHC experiments were carried out routinely. IHC staining scores were as follows: The scoring standards of the staining intensity were: 0 (no staining), 1 (light yellow staining), 2 (brown and yellow staining) and 3 (brown staining). The scoring standards for the percentage of positive cells under the microscope were: 0 (<5%), 1 (5–25%), 2 (25–50%), 3 (51–80%), and 4 (>80%). The final score was obtained by multiplying the stain intensity score by the positive cell percentage score, where <7 was low and $\geq 7$ was high.

## Bioinformatics analysis

The comparison of PRR14 expression level in colon cancer tissues and normal tissues was performed using the GEPIA database (http://gepia.cancer-pku.cn) [12] and the Oncomine database (https://www.oncomine.org) [13]. The Hong Y Dataset (GSE9348) of Oncomine was used [14]. The patient survival data of TCGA was obtained from the Human Protein Atlas database (https://www.proteinatlas.org). The optimal cut-off value for PRR14 expression level was determined using SPSS ROC curve analysis. The survival curve was estimated using Kaplan-Meier analysis, and the *P*-values were calculated with the log-rank (Mantel-Cox) test.

## siRNA synthesis and transfection

The siRNA against PRR14 and negative control (NC) siRNA were synthesized by Shanghai GenePharma Co. (Shanghai, China). The SiRNA sequences were referred to a previous study [3]. The sense sequence of the siRNA for PRR14-1 was: `5'-GCU AGA AGA UGU CAU GGC UTT-3'`, and the antisense sequence for PRR14-1 was: `5'-AGC CAU GAC AUC UUC UAG CTT-3'`. The sense sequence for PRR14-2 was: `5'-GGA CUG CCU CGA CCA AUC ATT-3'`, and the antisense sequence for PRR14-2 was: `5'- UGA UUG GUC GAG GCA GUC CTT-3'`. The sense sequence for the negative control was: `5'-UUC UCC GAA CGU GUC ACG UTT-3'`, and the antisense sequence for the negative control was: `5'- ACG UGA CAC GUU CGG AGA ATT-3'`. The siRNA was transfected into the cells at 30nM concentration using Lipofectamine RNAiMAX reagent (13778–150, invitrogen) according to the manufacturer's instructions. Cells were collected and analyzed 48h after transfection.

## RNA isolation and RT-PCR

Total RNA was isolated using RNA-simple Total RNA kit (DP419, Tiangen Biotech Co., Beijing, China) according to the manufacturer's instructions. Reverse transcription was done using ReverTra Ace qPCR RT kit (FSQ-101, Toyobo Co., Osaka, Japan). The RT-PCR assay was performed using SYBR Green PCR Master Mix (QPK-212, Toyobo Co.) on the CFX96 real-time PCR detection system (Bio-Rad, USA). The gene-specific primer sequences are listed in S1 Table. GAPDH was used for quantitative internal referencing. The $2^{-\Delta\Delta Ct}$ method was used to analyze the results.

## Western blot

After 48h of siRNA transfection, the cells were collected and lysed with T-PER™ Tissue Protein Extraction Reagent (Thermo Scientific, USA) containing protease inhibitor and phosphatase inhibitor (Roche, Mannheim, Germany). The protein concentration was determined with

the BCA kit (Thermo Scientific). The western blot experiment was carried out in accordance with the routine experimental steps. The following primary antibodies were used: the rabbit anti-PRR14 antibody (1:500, HPA060265, Sigma, USA), rabbit anti-CDK2 antibody (1:1000, SAB4300388, Sigma), rabbit anti-P21 antibody and P27 antibody (1:1000, 9932T, Cell Signaling technology, USA), rabbit anti-E cadherin antibody (1:200, BM3903, Boster, China), rabbit anti-N cadherin antibody (1:500, RLT2988, Ruiying Biological, China), mouse anti-Vimentin antibody(1:500, 60330-1-IG, Proteintech, China), rabbit anti-twist1 antibody (1:1000, SAB2108515, Sigma), rabbit anti-AKT antibody (1:1000, 4691T, Cell Signaling technology), rabbit anti-phospho-AKT$^{(Thr308)}$ antibody (1:1000, 13038T, Cell Signaling technology), rabbit anti-phospho-AKT$^{(Ser473)}$ antibody (1:1000, 4060T, Cell Signaling technology), mouse anti-GAPDH antibody (1:1000, AF0006, Beyotime, China). The secondary antibody used (goat anti-rabbit or goat anti-mouse secondary antibody) was conjugated to horseradish peroxidase (1:2000; Beyotime). Chemiluminescence pictures were taken using the Super Signal Sensitivity Substrate Kit (Pierce, Thermo Scientific) with the Chemi Doc XRS imaging system (Bio-Rad, USA).

## Cell proliferation assay

Cell proliferation was measured using the CCK8 reagent (HY-k0301, MedChemExpress, USA). Briefly, cells transfected with siRNA for 24 hours were counted and placed into 96-well plates with 1500 cells per well. 100μl DMEM containing 10% CCK8 reagent was added into the cells per well at 0h, 24h, 48h, and 72h, respectively. The cells were incubated at 37 $^{o}$C for 1h in a humidified atmosphere of 5% $CO_2$. Absorbance was measured at a wavelength of 450nm by using the Varioskan Flash microplate reader (Thermo Scientific, USA).

## Cell cycle and apoptosis analysis

For cell cycle analysis, cells were centrifuged and collected 48h after transfection, washed once with cold PBS, then fixed with 70% ethanol at 4 $^{o}$C refrigerator overnight. The fixed cells were centrifuged and collected the next day, washed twice with cold PBS, stained with PI containing RNase (c1052,Beyotime Biotechnology, China), and incubate at 37 $^{o}$C in the dark for half an hour. Apoptosis detection was carried out using the FITC Annexin V Apoptosis Detection Kit with PI (640914, Biolegend, USA). Cells were washed twice with cold BioLegend's Cell Staining Buffer (Biolegend), and then resuspended in Annexin V Binding Buffer at a concentration of 1.0 x10$^6$ cells/ml. 100μl of the cell suspension was transferred into a 1.5ml test tube. 5μl of FITC Annexin V, followed by 10μl of Propidium Iodide Solution, was added to the cell suspension. The cells were gently vortexed and incubated in the dark for 15min at room temperature. 400μl of Annexin V Binding Buffer was added to each tube. The apoptotic samples were detected with flow cytometry (BD Biosciences, USA).

## Tumor xenografts

Female nude mice were maintained in pathogen free conditions and used at 5–6 weeks of age. Animal ethics was approved by the Laboratory Animal Center of Chongqing Medical University. The mice were anesthetized with isoflurane controlled by small animal anesthesia machine. The siNC and siPRR14 HCT116 cells suspension (5 x 10$^5$ cells in 100ul PBS per point) were injected with insulin syringe into the left and right dorsal flank of the mice, respectively. About 30 days after injection, the mice were sacrificed by tearing the cervical vertebra and dissected, tumor samples were taken, and the length, width and weight of the tumor were measured. The volume was calculated by using the modified ellipsoid formula: volume = 1/2 (length x width$^2$).

## Wound healing, migration and invasion assays

For wound healing assay, the cells were digested and counted. 1 x 10$^6$ cells were seeded into each well of a 24-well plate, and incubated overnight. The cells were scratched in the middle of the plate with white pipette tips, washed with PBS for 3 times to remove the floating cells, and then photographed at 0h, 24h and 48h respectively. Transwell Chambers with multiporous 8μm pore size PET membranes (MCEP24H48, Millipore, Germany) were used for the migration and invasion assays. 2 x 10$^5$ cells were resuspended in 200μl serum-free medium and inoculated into the upper chamber. 500ml medium containing 10% FBS was added into the lower chamber. The cells were incubated at 37°C in a humidified atmosphere of 5% $CO_2$ for 24h, and then fixed with polyformaldehyde for 20min. Thereafter, they were washed 3 times with PBS, stained with crystal violet for 10min, and washed 3 times with PBS. The cells in the upper chamber were then gently wiped off with cotton swab, and the cells in the lower chamber were photographed.

## Immunofluorescent assay

For Brdu assay, 48h after siRNA transfection, cells were incubated with Brdu (ab142567, abcam, USA) for 1h, fixed with 4% paraformaldehyde for 20min, permeabilized with 0.3% triton for 15min, treated with 2M HCl for 20min, balanced with sodium tetraborate for 5min, and then blocked with 5% BSA for 1h. After fixation, cells were incubated with rabbit anti-PRR14 antibody (1:200,HPA060265,Sigma) or rat anti-Brdu antibody (1:300,ab6326,abcam, USA) at 4 °C overnight, followed by incubation with the secondary antibody at room temperature for 1 hour. For phalloidin staining, cells were fixed and permeabilized, then stained with TRITC-Phalloidin (40734ES75, Yeasen, China) for 1h at room temperature. The nuclei were stained with DAPI for 10min, and the Zeiss LSM 780 laser confocal microscope was used for photography.

## Statistical analysis

Results are shown as mean ± SD. Statistical evaluation was performed with SPSS. The student $t$ test was used to determine differences between groups. The $\chi^2$ test was used to examine the relationship between the protein expression of PRR14 and the clinicopathological factors. Statistical significance was set at $^{**}P<0.01$ and $^*P<0.05$. Graphs were prepared using Graphpad Prism 6.0.

# Results

## PRR14 is highly expressed in colon cancer, and this correlates with poor prognosis of the disease

Firstly, we used the GEPIA database to investigate the expression pattern of PRR14 in colon cancer. The data showed that the mRNA expression of PRR14 was higher in the cancer tissues than in the normal tissues. PRR14 expression was significantly correlated with tumor grade (Fig 1A and 1B). The data from oncomine database also demonstrated a higher expression of PRR14 in tumor tissues (Fig 1C). The TCGA survival data from the human Protein Atlas database showed that higher PRR14 expression was associated with worse patient outcomes (Fig 1D). We also evaluated PRR14 expression by using tissue microarray from clinical patients. PRR14 positive signal was mainly detected in the nucleus of non-neoplastic epithelium cells and tumor cells, and PRR14 staining was obviously stronger in the cancer tissues than in the adjacent normal tissues (Fig 1E and 1F). The results of the comparison of the expression of PRR14 among the clinicopathological features of colon cancer is displayed in Table 1. The

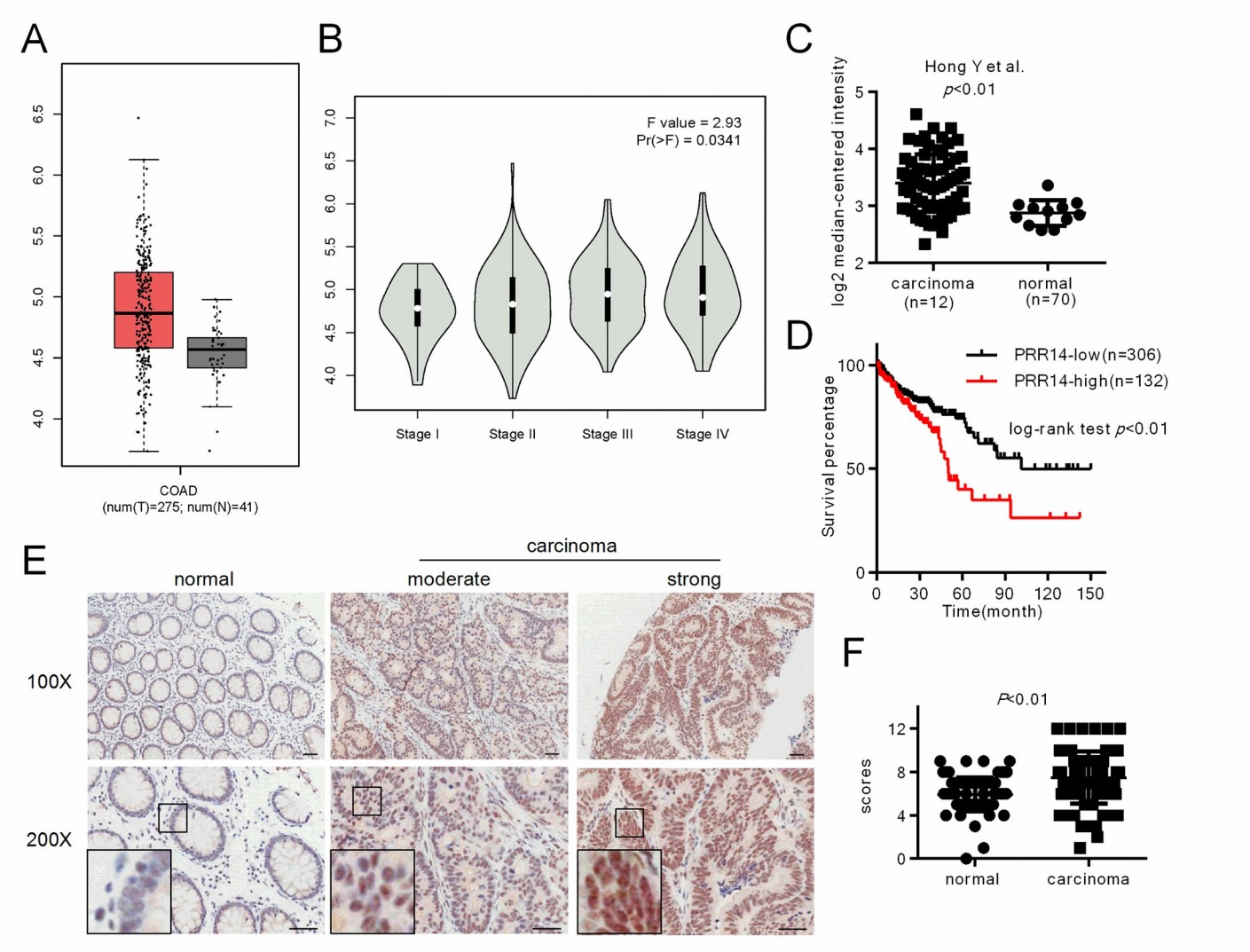

**Fig 1. PRR14 expression in colon cancer.** (A) Expression of PRR14 in cancer tissues and their corresponding adjacent normal tissues was analyzed from the online GEPIA database. (B) PRR14 expression in different grades of colon cancer samples was analyzed from the GEPIA database. (C) Expression of PRR14 in cancer tissues and adjacent normal tissues was analyzed from the online oncomine database. (D) Survival curve of the overall survival rate with High PRR14 and Low PRR14 expression was analyzed from the TCGA data of the Human Protein Atlas database. (E) Representative immunohistochemistry images of PRR14 expression in colon cancer tissues and their corresponding adjacent normal tissues. (F) Immunohistochemistry scores of PRR14 expression in the colon cancer tissues were significantly higher compared to the corresponding adjacent normal tissues. Scale bar, 100μm.

table shows that PRR14 expression in colon cancer was strongly associated with tumor size ($P = 0.012$), distant metastasis ($P = 0.045$) and TNM stage (Tumor Node Metastasis stage) ($P = 0.029$), but not with the gender and age of the patients.

## Knockdown of PRR14 inhibited cell growth *in vitro* and *in vivo*

To investigate the biological function of PRR14 *in vitro*, we knocked down PRR14 in two colon cancer cell lines–HCT116 and RKO–via siRNA-mediated gene silencing. The knockdown efficiency was determined by RT-PCR and western blot. The PCR results showed that the two siRNAs had a knockdown effect on the mRNA levels (Fig 2A), while the western blot results showed that siRNA-2 had a better knockdown effect at the protein level (Fig 2B).

**Table 1. The relationship between PRR14 expression levels and clinicopathological characteristics of colon cancer patients.**

| Clinicopathological features | | Number of patients | PRR14 expression status | | P |
|---|---|---|---|---|---|
| | | | Low(n = 39)(%) | High(n = 41)(%) | |
| Gender | Male | 43 | 20 (46.5%) | 23 (53.5%) | 0.617 |
| | Female | 37 | 19 (51.4%) | 18 (48.6%) | |
| Age | ≥60 | 55 | 27 (49.1%) | 28 (50.9%) | 0.928 |
| | <60 | 25 | 12 (48%) | 13 (52%) | |
| Tumour size | ≥6cm | 34 | 11 (32.4%) | 23 (67.6%) | 0.012 |
| | <6cm | 46 | 28 (60.9%) | 18 (39.1%) | |
| Lymph node metastasis | negative | 54 | 30 (55.6%) | 24 (44.4%) | 0.079 |
| | positive | 26 | 9 (34.6%) | 17 (65.4%) | |
| Distant metastasis | negative | 76 | 39 (51.3%) | 37 (48.7%) | 0.045 |
| | positive | 4 | 0 (0%) | 4 (100%) | |
| TNM stage | I、II | 52 | 30 (57.7%) | 22 (42.3%) | 0.029 |
| | III、IV | 28 | 9 (32.1%) | 19 (67.9%) | |

$\chi^2$ test was used for comparison between PRR14$^{low}$ and PRR14$^{high}$ tissues.

Therefore, siRNA-2 was selected for the following cell experiments. Immunofluorescence results also showed that the knockdown effect of siRNA-2 was obvious, and the enlarged figure showed that PRR14 expression was mainly located in the nuclear membrane and nucleus of cells, with a small amount of staining in the cytoplasm (Fig 2C).

We analyzed the effect of PRR14 on colon cancer cell growth using the CCK8 assay. The results showed that the cell proliferation rate was decreased in both HCT116 and RKO cells after transfection with the PRR14 siRNA (Fig 2D and 2E). Meanwhile, PRR14 did not affect the rate of apoptosis in both cell lines (S1 Fig). Furthermore, colony formation assay showed that PRR14 knockdown significantly reduced the clonogenic ability of the cells (Fig 2F). Subcutaneous tumorigenesis experiment showed that the tumor volume and weight were significantly lower in the knockdown group than in the control group (Fig 2G and 2H). These results show that PRR14 plays a role in the growth of colon cancer cells.

## Knockdown of PRR14 resulted in cell cycle arrest at G1 phase

To test the mechanism by which PRR14 affects proliferation, we examined the cell cycle of cells after RNA interference. Results showed that knockdown PRR14 resulted in cell arrest at G1 phase and a significantly lower proportion of cells at S phase (Fig 3A). Brdu labeling assay also indicated that the proportion of S-phase cells decreased (Fig 3B).

## Knockdown of PRR14 inhibited cell migration and invasion

The scratch-wound assay and transwell assay were performed to assess the effects of PRR14 on migration and invasion of HCT116 and RKO cells. The scratch assay showed that the wound healing time in the PRR14 siRNA group was significantly slower than in the control siRNA group (Fig 4A). The transwell experiment and matrigel coated transwell experiment showed that the migration and invasion ability of the siRRR14-treated cells were decreased (Fig 4B). This suggest a role of PRR14 in the migration and invasion of colon cancer.

## Knockdown of PRR14 affected the expression of EMT-related genes and the formation of pseudopodia

Interference with PRR14 in mesenchymal-like HCT116 and RKO cells resulted in an obvious shift from scattered spindle-shaped cells to clustered round-shaped cells (Fig 5A). We found

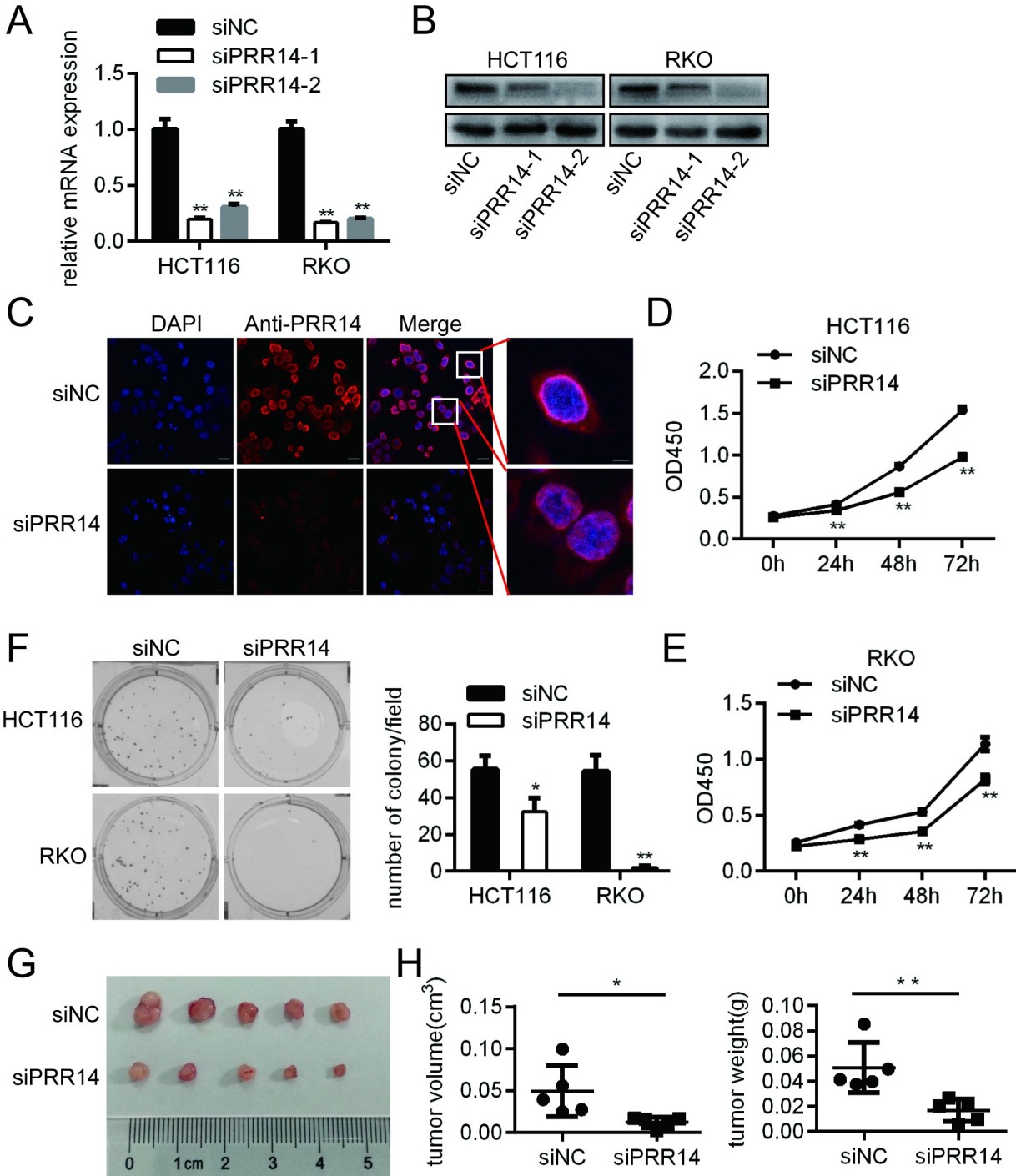

**Fig 2. Colon cancer cells proliferation following PRR14 depletion.** (A&B) The efficiency of siRNA interference was analyzed by (A) qRT-PCR and (B) Western blot assays. (C) Immunofluorescence assay of PRR14 expression in HCT116 cells. (D&E) The proliferation of HCT116 and RKO cells following PRR14 depletion. (F) PRR14 depletion inhibited colony formation of HCT116 and RKO cells. (G) Representative images of nude mouse subcutaneous tumors from HCT116 cells transfected with siNC and siPRR14. (H) The volume and the weight of the tumors were measured (*: $P < 0.05$; **: $P < 0.01$).

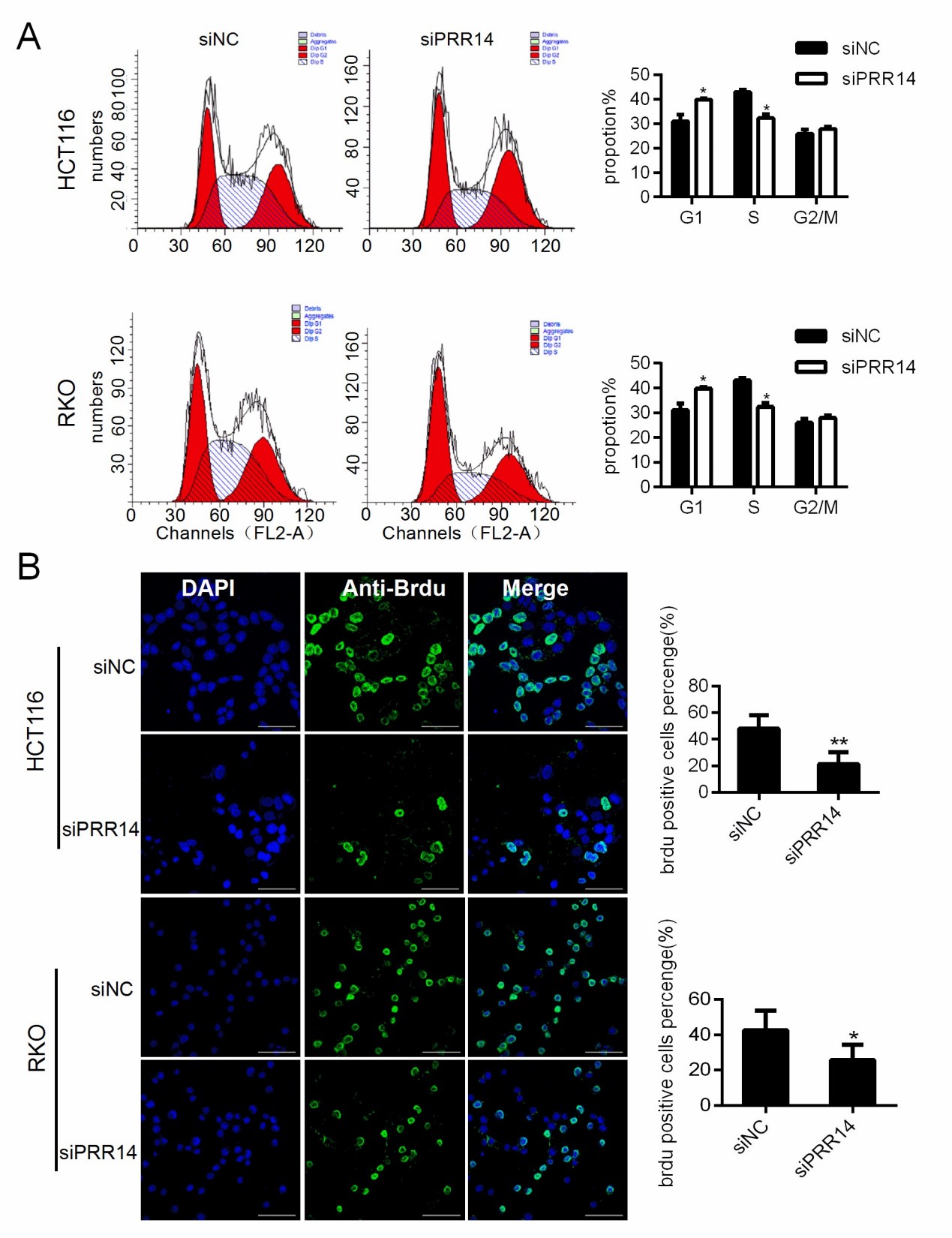

**Fig 3. The impact on cell cycle progression of colon cancer cells following PRR14 depletion.** (A) Cell cycle analysis in HCT116 and RKO cells transfected with siNC and siPRR14 siRNA by flow cytometry,and the average percentage of G0/G1,S,G2/M phases was calculated. (B) Representative immunofluorescence images of Brdu staining of HCT116 and RKO cells transfected with siNC and siPRR14 siRNA. The number of Brdu labeled positive cells was calculated to reflect the proportion of S-phase cells (*:$P < 0.05$;**: $P< 0.01$). scale bar,50um.

that knockdown of PRR14 inhibited EMT, as evidenced by the up-regulation of E-cadherin expression and the down-regulation of N-cadherin, Vimentin, and Twist1 expressions (Fig 5B). Further, we examined the protein expression of PRR14 and EMT makers in the different xenografts. We pooled tumors of two groups (No.9 and 10) since their small size. Western blot results and statistics analysis showed that the mesenchymal indicator Twist1 and Vimentin were highly expressed in xenografts with control cells, and the epithelial indicator E-cadherin was generally highly expressed in xenografts with PRR14 knockdown cells (Fig 5C and 5D). By immunofluorescent labeling of actin filaments, we found that the number of pseudopodia in the cells was significantly reduced (Fig 5E). These results demonstrate that PRR14 may affect colon cancer metastasis by regulating EMT and cytoskeletal remodeling.

## Knockdown of PRR14 affected the expression of cell cycle-related genes and AKT pathway genes

To clarify the mechanism by which PRR14 regulates the cell cycle, we knocked down PRR14 and examined its effects on the expression of cell cycle regulatory genes. Knockdown of PRR14 in HCT116 and RKO cells significantly inhibited CDK2 mRNA expression and protein expression (Fig 6A, 6B and 6C). In HCT116 cells, the mRNA level of CDK1, CCNA, CCNB, CCND1 and CDK kinase inhibitors–P21 and P27 were found to be elevated, while no significant change in RKO cells (Fig 6A and 6B). We also found that P21 and P27 protein level was elevated both in HCT116 and RKO cells (Fig 6C). Moreover, knockdown of PRR14 inhibited the phosphorylation of AKT and its downstream gene, mTOR (Fig 6D). These results suggest that PRR14 modulates the cell cycle gene expression and acts on the AKT pathway during colon cancer progression.

## Discussion

In this study, we found that PRR14 was significantly upregulated in colon cancer, and this is consistent with the results of previous analysis of the TCGA database [9]. PRR14 expression was associated with the malignant clinicopathological characteristics, including tumor size, distant metastasis and TNM stage of colon cancer. SiRNA-mediated gene silencing of PRR14 inhibited colon cancer cell proliferation, cell cycle progression, migration and invasion. This is in line with similar reports made about this gene in lung cancer cells and breast cancer cells [8]. Thus, these findings indicate a role of PRR14 in promoting tumor or cancer progression.

Malignant proliferation of tumor cells is associated with dysregulation of their cell cycle progression [15]. Cell cycle progression is precisely regulated by a series of periodic factors [16, 17]. We found that knockdown of PRR14 inhibited the expression of CDK2, a G1/S transition promoting gene, in both cell lines. Moreover, in HCT116 cells, the expressions of CDK1, CCNA, CCNB and CCND1 were also down-regulated and the expressions of CDK2 inhibitors–P21 and P27 were up-regulated, while in RKO cells, the expression of P21 and P27 protein levels were up regulated. These results suggest that PRR14 may promote malignant proliferation of colon cancer cells by disrupting cell cycle progression. Besides, the inconsistent mRNA levels of cell cycle genes in two cell lines indicating that the mechanism of PRR14 action may be inconsistent between the two cells in mRNA level. As the mRNA and protein of

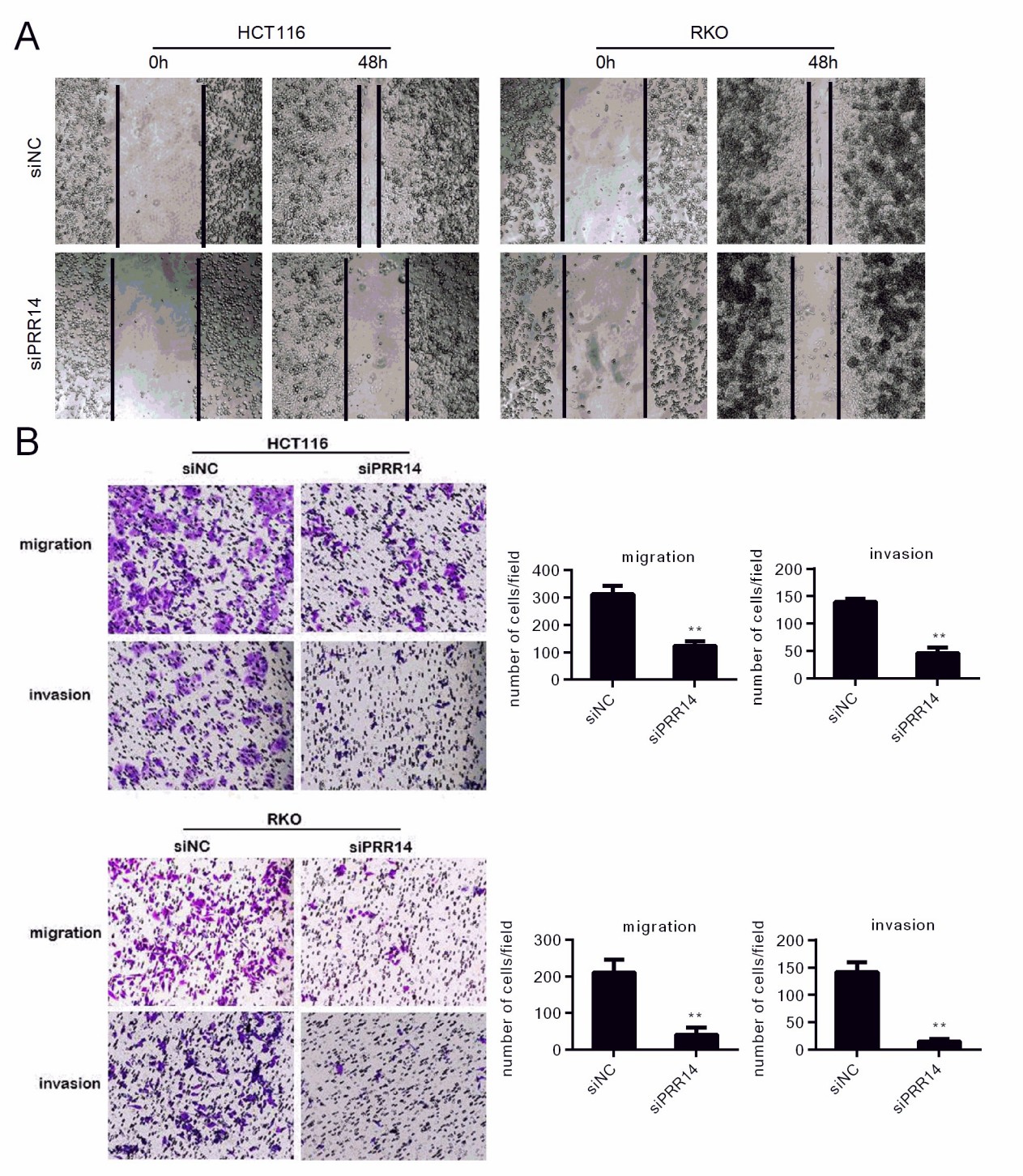

**Fig 4. The impact on migration and invasion of colon cancer cells following PRR14 depletion.** Cell migration or invasion was evaluated by using (A) would healing assay (B) transwell migration and invasion assay in HCT116 and RKO cells, respectively. (**: $P < 0.01$).

P21 and P27 are not consistent in RKO cells, we speculate that PRR14 may play a more important role in the post-translation level.

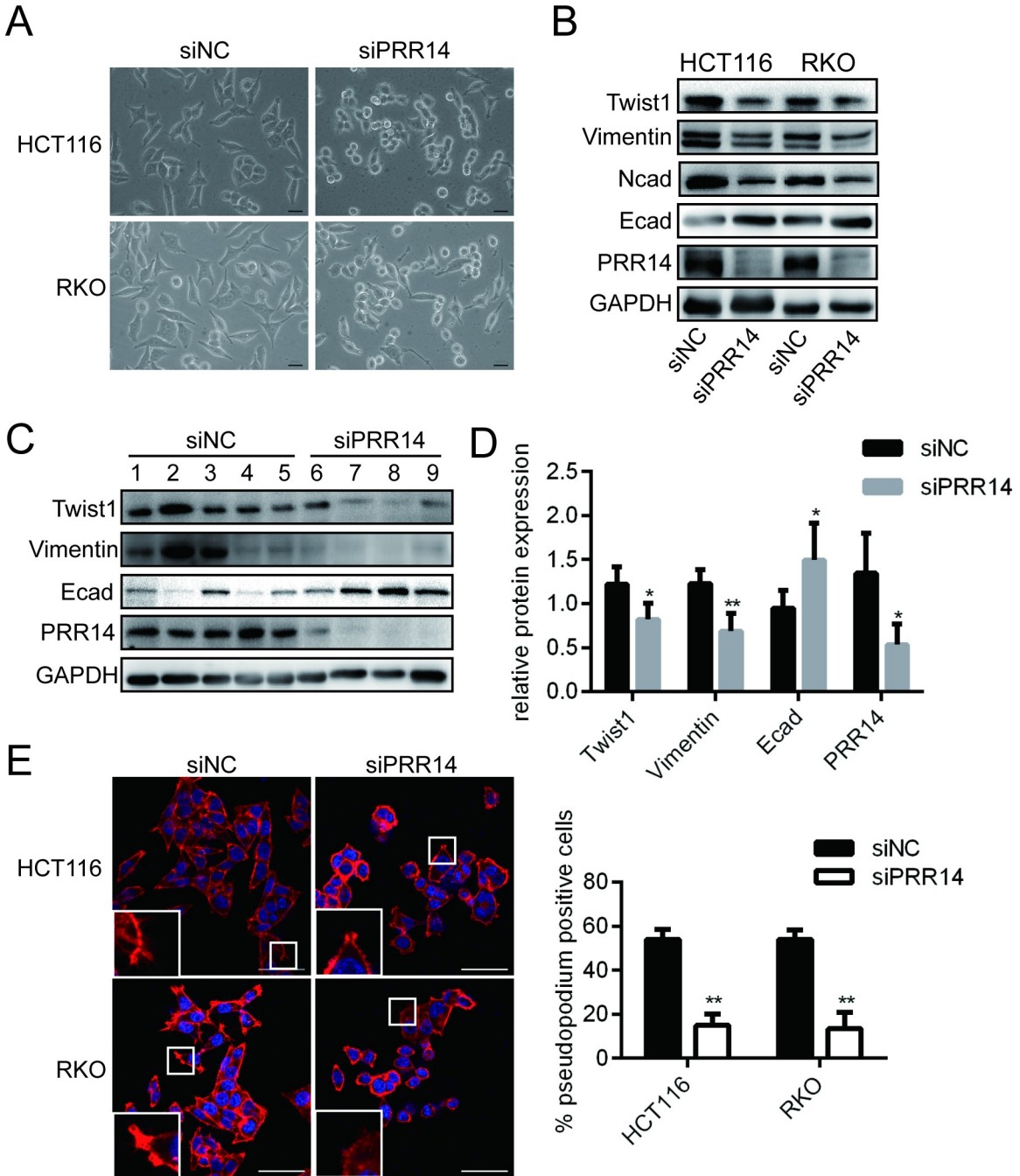

**Fig 5. Modulation of EMT and the formation of pseudopodia in the colon cancer cells following PRR14 depletion.** (A) Morphologic changes in mesenchymal HCT116 and RKO cells after transfection with siNC and siPRR14. Scale bar, 50 μm. (B) Western blot was used to detect the expression of genes controlling EMT in HCT116 and RKO cells transfected with siNC and siPRR14. (C) Western blot detection of the PRR14 and EMT indicators expression in xenograft tumor cells. (D) Statistics comparison of PRR14 and EMT indicators protein level in the xenografts. (E) Representative immunofluorescence images of phalloidin staining of HCT116 and RKO cells transfected with siNC and siPRR14, and the calculated proportion of pseudopod-positive cells. scale bar, 50 μm. (**: $P < 0.01$).

Numerous evidences show that cell phenotype and movement are related to EMT. EMT is considered to be a critical step in the process of metastasis in a variety of tumors [18, 19], including colon cancer [20]. It is a complicated process in which cells are transformed from

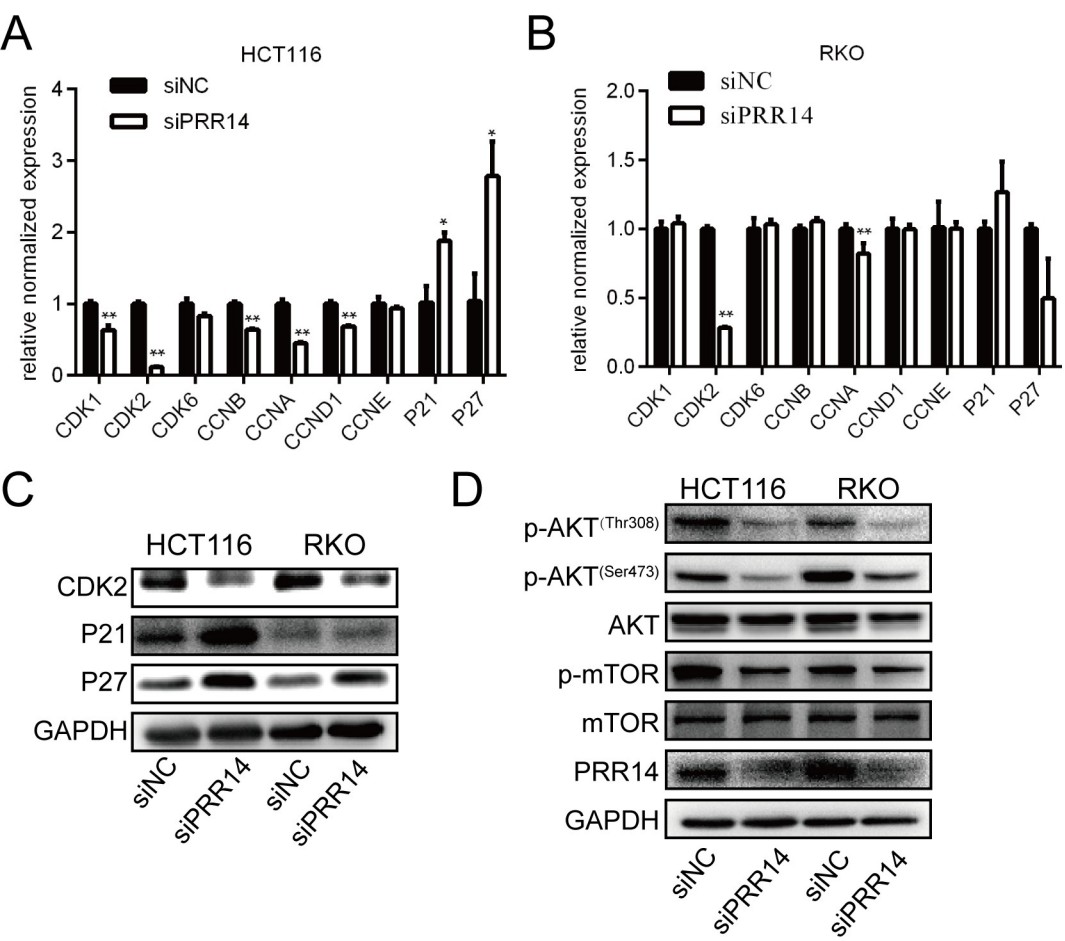

**Fig 6. Effects of PRR14 depletion on the expression of genes regulating the cell cycle and the related pathway.** (A&B) qRT-PCR was used to detect the cell cycle genes expression in HCT116 (A) and RKO cells (B) transfected with siNC and siPRR14. (C) Western blot was used to detect the cell cycle genes expression in HCT116 and RKO cells transfected with siNC and siPRR14. (D) Western blot was used to detect the AKT pathway gene expression in HCT116 and RKO cells transfected with siNC and siPRR14 siRNA. (*: $P < 0.05$; **: $P < 0.01$).

epithelial phenotype to mesenchymal phenotype, thus gaining migration ability. In this study, silencing of PRR14 changed the cells' morphology, and decreased the expression of Twist1, an important transcription factor in the EMT process, which mediates the inhibition of E cadherin expression and the up-regulation of N cadherin expression. This indicates that PRR14 is involved in the EMT process. Since the cytoskeleton enhances pseudopodia formation during the motility of cancer cells [21–24], and PRR14 was found to regulate cytoskeletal remodeling, it could be stated that PRR14 promotes the metastasis of colon cancer cells by modulating the structure of the cytoskeleton.

The PI3K/AKT pathway is involved in the regulation of proliferation, apoptosis and migration in many tumors, including colon cancer. We have shown that silencing PRR14 induced changes in p-AKT and p-mTOR levels, indicating that PRR14 acts on PI3K/AKT pathway. This is consistent with observations made about lung cancer cells. It has been reported that PRR14 activates the PI3K/AKT pathway by binding with the SH3 domain of GRB2 [8]. GRB2 is widely expressed in a variety of cells and participates in many cellular functions by transducing the receptor tyrosine kinase (RTK) signaling pathway. The SH2 domain of GRB2 receives RTK signals on the cell membrane and mediates the signals to be transmitted to the

downstream RAS/MAPK and PI3K/AKT pathways[25–27]. A previous study has shown that PRR14 functions in the nucleus to regulate the localization of heterochromatin [3]. How it binds to GRB2 in the cytoplasm, or whether it has other variants involved in GRB2 binding in the cytoplasm, is worth studying. Apart from acting on the PI3K pathway through GRB2, whether PRR14 has another mechanism through which it regulates the development and progression of colon cancer, remains to be determined. It has been reported that PRR14's proline-rich domain is involved in protein-protein interaction by binding to multiple domains [4, 5]. This suggests that PRR14 may interact with other unknown molecules in regulating colon cancer development and progression, and the identification of these molecules is our next research direction.

## Conclusion

PRR14 plays an important role in the development and progression of colon cancer, and is a possible prognostic marker and therapeutic target for the disease. Further exploration to elucidate its specific molecular mechanism is of great significance.

## Supporting information

**S1 Fig. Effect of PRR14 depletion on the apoptosis of HCT116 and RKO cells.** Representative images of apoptosis detection, and the proportion of the sum of early apoptosis and late apoptosis.
(TIF)

**S1 Table. Genes-specific PCR primers.**
(DOC)

## Acknowledgments

We are grateful to Enoch Appiah Adu-Gyamfi for proofreading and editing the document.

## Author Contributions

**Conceptualization:** Fangfang Li, Chundong Zhang, Lijuan Fu.

**Data curation:** Fangfang Li, Chundong Zhang, Lijuan Fu.

**Formal analysis:** Fangfang Li, Chundong Zhang, Lijuan Fu.

**Funding acquisition:** Fangfang Li, Chundong Zhang, Lijuan Fu.

**Investigation:** Fangfang Li, Chundong Zhang, Lijuan Fu.

**Methodology:** Fangfang Li, Chundong Zhang, Lijuan Fu.

**Project administration:** Fangfang Li, Chundong Zhang, Lijuan Fu.

**Resources:** Fangfang Li, Chundong Zhang, Lijuan Fu.

**Software:** Fangfang Li, Chundong Zhang, Lijuan Fu.

**Supervision:** Chundong Zhang, Lijuan Fu.

**Validation:** Fangfang Li, Chundong Zhang, Lijuan Fu.

**Visualization:** Fangfang Li, Chundong Zhang, Lijuan Fu.

**Writing – original draft:** Fangfang Li, Chundong Zhang, Lijuan Fu.

**Writing – review & editing:** Fangfang Li, Chundong Zhang, Lijuan Fu.

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
