## [Decision Letter · Decision Letter 0]

28 Jun 2019

PONE-D-19-15816

PRR14 Overexpression Promotes Cell Growth, Epithelial to Mesenchymal Transition and Metastasis of Colon Cancer via the AKT Pathway

PLOS ONE

Dear Prof Fu,

Thank you for submitting your manuscript to PLOS ONE. After careful consideration, we feel that it has merit but does not fully meet PLOS ONE’s publication criteria as it currently stands. Therefore, we invite you to submit a revised version of the manuscript that addresses the points raised during the review process.

Two experts have reviewed the currrent manuscript and found that the study was interesting and written well but there more aspects to be expalined by further editorial and experimental works enough for a publication in the journal.

We would appreciate receiving your revised manuscript by Aug 12 2019 11:59PM. To enhance the reproducibility of your results, we recommend that if applicable you deposit your laboratory protocols in protocols.io, where a protocol can be assigned its own identifier (DOI) such that it can be cited independently in the future. For instructions see: http://journals.plos.org/plosone/s/submission-guidelines#loc-laboratory-protocols

We look forward to receiving your revised manuscript.

Kind regards,

Jung Weon Lee, Ph.D.

Academic Editor

PLOS ONE

**Journal Requirements:**

2. To comply with PLOS ONE submissions requirements, in your Methods section, please provide additional information on the animal research and ensure you have included details on (1) methods of sacrifice, (2) methods of anesthesia and/or analgesia, and (3) efforts to alleviate suffering.

**Comments to the Author**

1. Is the manuscript technically sound, and do the data support the conclusions?

Reviewer #1: Yes

Reviewer #2: Yes

2. Has the statistical analysis been performed appropriately and rigorously? 

Reviewer #1: Yes

Reviewer #2: Yes

3. Have the authors made all data underlying the findings in their manuscript fully available?

Reviewer #1: Yes

Reviewer #2: Yes

4. Is the manuscript presented in an intelligible fashion and written in standard English?

Reviewer #1: Yes

Reviewer #2: Yes

5. Review Comments to the Author

Reviewer #1: The authors investigated PRR14 (proline-rich protein 14) in colon cancer cells. Using multiple databases and a colon cancer tissue microarray, Fig. 1 shows that PRR14 is more highly expressed in colon cancer compared to non-cancer colon tissue. Higher PRR14 is correlated with lower survival. Using HCT116 and RKO colon cancer cell lines, Fig. 2 shows that siRNA knockdown of PRR14 slows proliferation and reduces tumor growth in xenografted mice. Fig. 3 shows PRR14 knockdown reduces invasion. Fig. 4 shows that reduction of PRR14 increases E-cadherin and reduces mesenchymal markers twist, vimentin, and N-cadherin and pseudopodia formation. In Fig. 5, PRR14 knockdown lowers some cell cycle markers and phosphor-AKT/mTOR. The authors conclude that PRR14 plays an important role in colon cancer and may possibly be used as a biomarker.

The literature on PRR14 is very little so not much is known on its function. Its potential role in colon cancer is suggested but the authors should be less emphatic that it has an important role. Some suggestions to help improve the quality of the paper is as follows:

1. What is cellular localization of PRR14 in HCT/RKO cells? If predominantly nuclear, difficult to explain results. IHC in Fig. 1E is difficult to determine localization (higher mag as insert would help).

2. What is effect of PRR14 knockdown on other signaling pathways such as ERK or JNK. Does PRR14 have a global role in intracellular signaling?

3. What is the cell cycle profile with PRR14 (G1/S block?).

4. Further comment on differences in PRR14 kd in 2 cell lines. E.g., p21/p27 mRNA; despite no change p27 mRNA, protein increased in RKO.

5. Table 2 distant mets n=4 all with high PRR14 (100%), whereas low n=0 yet number in parenthesis is 48.8%; should be 0. How explain P=0.029 for TNM but 0.079 ln mets?

6. Tumor xenografts---presumably kd is transient if using siRNA. Level of PRR14 in tumors? How long after siRNA transfection does PRR14 kd last in vitro?

Reviewer #2: The manuscript “PRR14 Overexpression Promotes Cell Growth, Epithelial to Mesenchymal Transition

and Metastasis of Colon Cancer via the AKT Pathway” by Fangfang Li

et al., investigates the molecular mechanism through wich PRR14, a member of the proline-rich protein family

partecipates to tumor progression. The experiments are well conceived, the manuscript is well written and I have only few comments on it.

Major comments

HCT116 and RKO

mesenchimal one. What happened if PRR14 is overexpressed (or silenced) in epithelial colon cancer cells? Some keys experiments should be presented in epithelial colon cancer cells.

On the same way it would be interest to see the effect of PRR14 overexpression in HCT116 and RKO cells. Also if the results are negative they should be presented.

Minor comments

The sentence “To clarify the mechanism by which PRR14 regulates the cell cycle

” at the beginning of the result section “Knockdown of PRR14 affected the expression of cell cycle-related genes and AKT

pathway genes” is not correct. Indeed the authors did not investigate at all the ability of PRR14 to regulate cell cycle but they address the point of its role in proliferation

(not in specific cell cycle phases).

Figure 5C is not mentioned in the text

6. PLOS authors have the option to publish the peer review history of their article (what does this mean?). If published, this will include your full peer review and any attached files.

Reviewer #1: Yes: Carlos Perez-Stable

Reviewer #2: Yes: Giulia Piaggio

---

## [Author Response · Author response to Decision Letter 0]

8 Aug 2019

We have responded the comments point by point in the response letter, and revised the manuscript as per editor and reviewers’ comments and construction suggestions. Please kindly find them in the response letter and revised manuscript.

---

## [Decision Letter · Decision Letter 1]

26 Sep 2019

PRR14 Overexpression Promotes Cell Growth, Epithelial to Mesenchymal Transition and Metastasis of Colon Cancer via the AKT Pathway

PONE-D-19-15816R1

Dear Dr. Fu,

We are pleased to inform you that your manuscript has been judged scientifically suitable for publication and will be formally accepted for publication once it complies with all outstanding technical requirements.

With kind regards,

Jung Weon Lee, Ph.D.

Academic Editor

PLOS ONE

Additional Editor Comments (optional):

Reviewers' comments:

Reviewer's Responses to Questions

**Comments to the Author**

1. If the authors have adequately addressed your comments raised in a previous round of review and you feel that this manuscript is now acceptable for publication, you may indicate that here to bypass the “Comments to the Author” section, enter your conflict of interest statement in the “Confidential to Editor” section, and submit your "Accept" recommendation.

Reviewer #1: All comments have been addressed

Reviewer #2: All comments have been addressed

2. Is the manuscript technically sound, and do the data support the conclusions?

Reviewer #1: Yes

Reviewer #2: Yes

3. Has the statistical analysis been performed appropriately and rigorously? 

Reviewer #1: Yes

Reviewer #2: N/A

4. Have the authors made all data underlying the findings in their manuscript fully available?

Reviewer #1: Yes

Reviewer #2: Yes

5. Is the manuscript presented in an intelligible fashion and written in standard English?

Reviewer #1: Yes

Reviewer #2: Yes

6. Review Comments to the Author

Reviewer #1: (No Response)

Reviewer #2: All my comments have been addressed. The manuscript is now acceptable for publication on PLOSONE journal.

7. PLOS authors have the option to publish the peer review history of their article (what does this mean?). If published, this will include your full peer review and any attached files.

Reviewer #1: Yes: Carlos Perez-Stable

Reviewer #2: Yes: Giulia Piaggio

---

## [Editor Report · Acceptance letter]

30 Sep 2019

PONE-D-19-15816R1 

PRR14 Overexpression Promotes Cell Growth, Epithelial to Mesenchymal Transition and Metastasis of Colon Cancer via the AKT Pathway 

Dear Dr. Fu:

I am pleased to inform you that your manuscript has been deemed suitable for publication in PLOS ONE. Congratulations! Your manuscript is now with our production department. 

With kind regards,

on behalf of

Dr. Jung Weon Lee 

Academic Editor

PLOS ONE